# Applying Acoustic Scattering Layer Descriptors to Depict Mid-Trophic Pelagic Organisation: The Case of Atlantic African Large Marine Ecosystems Continental Shelf

Anne Mouget [1,2,*], Patrice Brehmer [1,3], Yannick Perrot [1], Uatjavi Uanivi [4], Ndague Diogoul [1,5], Salahedine El Ayoubi [6], Mohamed Ahmed Jeyid [7], Abdoulaye Sarré [5], Nolwenn Béhagle [1], Aka Marcel Kouassi [8] and Eric Feunteun [2,9]

1   IRD, Univ Brest, CNRS, Ifremer, Lemar, Délégation Régionale IRD Ouest France, 29280 Plouzané, France; patrice.brehmer@ird.fr (P.B.); yannick.perrot@ird.fr (Y.P.); diogoulndague@yahoo.fr (N.D.); nolwenn.behagle@gmail.com (N.B.)
2   UMR BOREA (Biologie des Organismes et Ecosystèmes Aquatiques), (MNHN, CNRS, SU, IRD, UCN, UA), Station Marine de Dinard, CRESCO, 35800 Dinard, France; eric.feunteun@mnhn.fr
3   Commission Sous Régional des Pêches (CSRP), SRFC, Dakar BP 25485, Senegal
4   Ministry of Fisheries and Marine Resources (MFMR), Swakopmund P.O. Box 912, Namibia; uatjavi.uanivi@mfmr.gov.na
5   Centre de Recherche Océanographique de Dakar Thiaroye (CRODT), Institut Sénégalais de Recherche Agricole (ISRA), Pôle de Recherche de Hann, Dakar BP 2241, Senegal; ablaysarrey@yahoo.fr
6   Institut National de Recherche Halieutique (INRH), Casablanca 20250, Morocco; s_elayoubi@hotmail.com
7   Institut Mauritanien de Recherche Océanographique et des Pêches (IMROP), Nouadhibou BP 22, Mauritania; moahtaje@yahoo.fr
8   Centre de Recherches Océanologiques (CRO), Abidjan BP V 18, Côte d'Ivoire; aka.marcel.kouassi@gmail.com
9   CGEL (Centre de Géo-Écologie Littorale), EPHE-PSL, 35800 Dinard, France
*   Correspondence: anne.mouget@mnhn.fr

**Abstract:** To identify key nonspecific organisational characteristics of the mid-trophic pelagic communities, which remain a challenge, we work with sound scattering layers (SSLs). Application was tested in the three African Atlantic Large Marine Ecosystems (AALMEs) to assess the utility of adapting and developing new acoustic variables. Our methodology allowed comparison between Large Marine Ecosystems (LMEs) based on 14 morphological, spatial and acoustic variables to characterize SSLs over time. These original variables were effective in monitoring and comparing the LMEs, and even allowed us to discriminate different organisations inside the Canary Current LME. Common traits identified for all AALMEs included the importance of the shallowest SSL. However, we identified an unexpected night-time pattern in SSL distributions in the Benguela Current LME which reflect a major difference in diel vertical migration mechanisms relative to other AALMEs. We also highlighted the predominance of elementary sampling unit (ESUs) with a single SSL and an unoccupied tiny layer close to the bottom, even if some ESUs presented up to six SSLs. Inter-annual changes in SSL organisation are highlighted by adaptation of original variables as the proportion of the water column occupied by SSLs and the relative importance of the shallowest SSL. SSL variables have been used mostly in deep water; here, we applied them on the continental shelf. SSL variables can be used to standardize the monitoring of marine ecosystems and capture change in spatial structure and function of mid-trophic pelagic marine ecosystems worldwide, even in data-poor areas where species identification of lower-trophic pelagic organism remains a challenge.

**Keywords:** sound-scattering layer; fisheries acoustics; ecosystem variables; diel vertical migration; Matecho; Canary Current Large Marine Ecosystem; Benguela Current Large Marine Ecosystem

## 1. Introduction

Fisheries acoustics is effective for better understanding marine [1,2] and freshwater [3] ecosystems. Acoustic sensors are routinely used in scientific studies and monitoring

programs [4–6]. For a long time, pelagic echo layers (or sound-scattering layers, SSLs) were considered as background noise when interpreting acoustics to study marine ecosystems [7]. Often, the acoustic reflections of fish schools had to be extracted or filtered from acoustic backscatter data to be analyzed properly [8,9].

Biological aggregations have been widely studied to understand what species aggregate, how and why aggregation form, and how we can study them [10]. There is a large variety of aggregations, from top predators to phytoplankton. In this study, we focus on SSLs, which can be produced by aggregations of a variety of pelagic species [11], such as aggregations of zooplankton [12,13], micronekton [14,15], gelatinous macronekton [16] and even phytoplankton [17–19]. These structures can be found in all oceans [20] and many studies address specific aspects of SSL, such as organisation and interactions [21] or their impact as biological pump [22]. Nonetheless, SSLs might react strongly to anthropogenic impacts and environmental fluctuations to create changes through SSL descriptors, mostly due to their short life cycle [23,24]. Some studies analyze SSLs to better understand a geographic area and their relationship with environmental data [25,26], with or without biological sampling. The early study of acoustic properties of SSLs started in the 1940s [20] and continue to today [27,28], especially with the use of narrow-band echo-sounding systems [29].

The present study takes advantage of previous studies and in particular on the work of Proud et al. [28], who worked on the standardization of metrics characterizing SSL. Our study advances and shows a new methodological approach to monitor Large Marine Ecosystems (LMEs) using SSLs without biological data. Diel Vertical Migration (DVM) is highly informational on vertical structuring and biocenoses function [26,28,30]. We used DVM in parts of the three African Atlantic Large Marine Ecosystems (AA LMEs) to demonstrate their utility to discriminate changes within an LME as well as differences and similarities between LMEs. We expect to observe DVM of type I [31] in the studied areas.

The aim of this study is to show the utility of our methodology based on SSLs in its ability to monitor and compare AA LMEs in the context of data-poor areas. The application on AA LME allows illustration of the efficiency of the method on a large dataset. Previous studies on these areas provided the ecological context [32–38]. We expect to observe differences within and between AA LMEs [33] applying our methodology. Interannual changes occur [35,39,40], so we can expect reported SSL changes over years. We compare SSL signals from AA LMEs to determine which variables might be useful for monitoring temporal change. To do this, we identified innovative SSL variables to describe SSL organization over the continental shelf of AA LMEs by first exploring spatial features using a dataset from several sea surveys carried out using the same research vessel and methods. Then, we analyzed the following: (i) temporal variability of SSL organization relative to short, diel-scale processes, i.e., to record DVMs of plankton, [41,42] and then (ii) document SSL's inter-annual variation.

## 2. Materials and Methods

### 2.1. Material: Annual Acoustics Survey in African Atlantic Large Marine Ecosystems

Acoustic data were recorded using a 38 kHz echosounder [2] from 10 to 500 m depth on board the RV Dr Fridtjof Nansen. We used an ES38-B transceiver, hull mounted at a depth of 5.5 m, with an absorption coefficient of 8.7 dB km$^{-1}$, a pulse length of 1.024 ms and a maximum used transmission power of 2000 watts [43]. The echosounder was calibrated following classic calibration procedures [44]. Several authors have shown that the acoustic density measured at 38 kHz is a good indicator of micronektonic abundance [45–47]; this frequency is currently used in fisheries acoustic studies. SSLs have already been usually studied in deeper sea, e.g., [48–50]. In this study, we consider data on the continental shelf, i.e., with a depth under 150 m, to monitor changes in areas less studied than the high sea.

We recorded data over the continental shelf of the AA LMEs (Figure 1): the Canary Current Large Marine Ecosystem (CCLME), the Guinea Current Large Marine Ecosystem (GCLME) and the Benguela Current Large Marine Ecosystem (BCLME) [51] (Figure 1,

Table 1). The CCLME extended from the Strait of Gibraltar (36° N, 5° W) to south of Guinea-Bissau (11° N, 16° W), with our study area encompassing coastal waters from 34° N, 7° W to 12° N, 17° W. This area was sub-divided into two ecologically distinct regions for sampling and analyses: North and South of Cape Blanc [52]. The area north of Cape Blanc exhibited continuous upwelling, while upwelling in the area to the south was seasonal [53]. We used data from 14 surveys in the CCLME from 1995 to 2015 totalling 99,788 nmi.

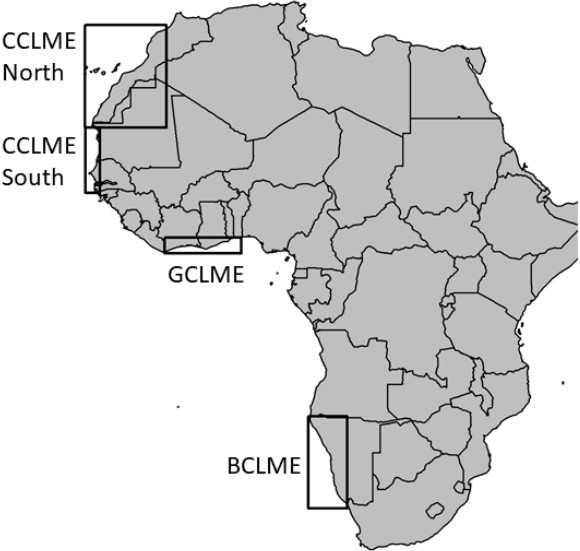

**Figure 1.** Geographic locations of the study areas sampled over the continental shelf of the African Atlantic Large Marine Ecosystems (LMEs). Rectangles represent study areas situated in the northern and southern Canary Current LME (CCLME), Guinea Current LME (GCLME) and Benguela Current LME (BCLME).

**Table 1.** Characteristics of acoustic surveys conducted in three African Atlantic Large Marine Ecosystems (LMEs). Annual surveys conducted in the Canary Current LME (n = 14) occurred in 1994–2006, 2011, 2015, in the Guinea Current LME (n = 8) in 1999-2006 and in the Benguela Current LME (n = 8) in 1994–2001. Transect lengths for each LME represent the total linear distance of all surveys conducted within the same LME; the Elementary Sampling Unit (ESU) is set at 0.1 nmi in length (during the same season and at depth < 150 m).

| LME | Geographic Position | Sampled Years | Transect (nmi) | Number of Analysed ESU |
|---|---|---|---|---|
| Canary Current | 34° N; 7° W to 12° N; 17° W | 1994–2006, 2011, 2015 | 96,788 | 588,459 |
| Canary Current North | 20.8° N; 7.2° W to 34.1° N; 17.7° W | 1994–2006, 2011, 2015 | 58,965 | 368,782 |
| Canary Current South | 12.2° N; 16.1° W to 20.8° N; 17.7° W | 1994–2006, 2011, 2015 | 37,822 | 219,677 |
| Guinea Current | 4° N; 8° W to 6° N; 3° E | 1999–2006 | 12,908 | 86,841 |
| Benguela Current | 17° S; 9° E to 31° S; 17° E | 1994–2001 | 39,368 | 1086 |

The GCLME extended from Bissagos Island (Guinea Bissau) in the north (11° N, 16° W) to Cape Lopez (Gabon) in the south (0° S, 8° E) [36]; our study area encompassed coastal waters from 4° N, 8° W to 6° N, 3° E. The dataset from the GCLME consisted of eight surveys from 1999 to 2006, totalling 12,908 nmi.

The BCLME occurs along the coast of south western Africa, stretching from the border between Namibia and Angola in the north (17° S, 11° W) southwards to the east of the Cape of Good Hope (South Africa) (29° S, 17° E) [37]. Our study area encompassed the coastal waters from 17° S, 9° E to 31° S, 17° E. Eight surveys were conducted in the BCLME from 1994 to 2001, totalling 39,368 nmi.

All survey designs were similar over years. Finally, we consider four marine ecosystems: CCLME North, CCLME South, GCLME and BCLME. A seasonal upwelling is present in both the southern Canary Current LME and the Guinea Current LME [34,54–56], whereas the upwelling in the northern Canary Current LME and the Benguela Current LME occurs year-round [53]. We used data from surveys not carried out during the seasonal upwelling.

### 2.2. Methods: Fisheries Acoustics and Sound Scattering Layers

Acoustic data were converted and cleaned using Matecho [57]. We used the echo-integrated data at a spatial resolution of 0.1 nmi by 1 m depth. To extract SSLs, we post-processed data with a segmentation algorithm within Matecho at an echo level threshold of $-70$ dB re 1 m$^{-1}$ (noted dB from here). To characterize the SSLs in the water column, we computed SSLs variables for each elementary sampling unit (ESU) of 0.1 nmi length. We used the $-70$ dB value as the lower threshold defining micronektonic SSLs [31,58]. Because there was no upper threshold, SSLs included zooplankton, micronekton and all other pelagic organisms. To assess SSL spatial pattern variability at the inter-annual level, we used mean volume backscattering strength ($S_v$ in dB) [6] as a proxy for relative pelagic abundance [58]. Before data analysis, we removed data related to transition periods in diel vertical migration patterns. Transition periods (dawn and dusk) were linked to sun azimuth data, i.e., by sun elevation, determined by data date, hour and geospatial position [59]. Hereafter, data from northern CCLME, southern CCLME, GCLME and BCLME were studied independently and named here after the respective marine ecosystems. Therefore, there are three LMEs and four marine ecosystems used in this study. We assumed that spatial auto-correlation in data along transects was negligible [14,60,61].

We used Matecho to compute six variables per ESU "j" for each SSL "i" (Figure 2): minimal depth ($đ_{i,j}$) [62], maximal depth ($Đ_{i,j}$) [62], width ($Ẇ_{i,j}$) [28,62], $S_v$ ($S_{v,i,j}$) [6], $s_A$ ($s_{A,i,j}$) [6] and the number of SSLs ($Ꞑ_j$) [30,62,63]. We focused on variables related to the shallowest SSL (i.e., $SSL_{1,j}$), and all SSLs, from the shallowest to the deepest ($SSL_{all,j}$). We choose to focus on the shallowest SSL because they to have a major importance in our marine ecosystems, according to expert experience. Presented variables come directly from Matecho, i.e., $đ_{1,j}$, $Đ_{1,j}$, $Ẇ_{i,j}$, $S_{v,1,j}$, $s_{A,1,j}$, and we calculated $đ_{all,j}$, $Đ_{all,j}$, $Ẇ_{all,j}$, $S_{v,all,j}$, $s_{A,all,j}$.

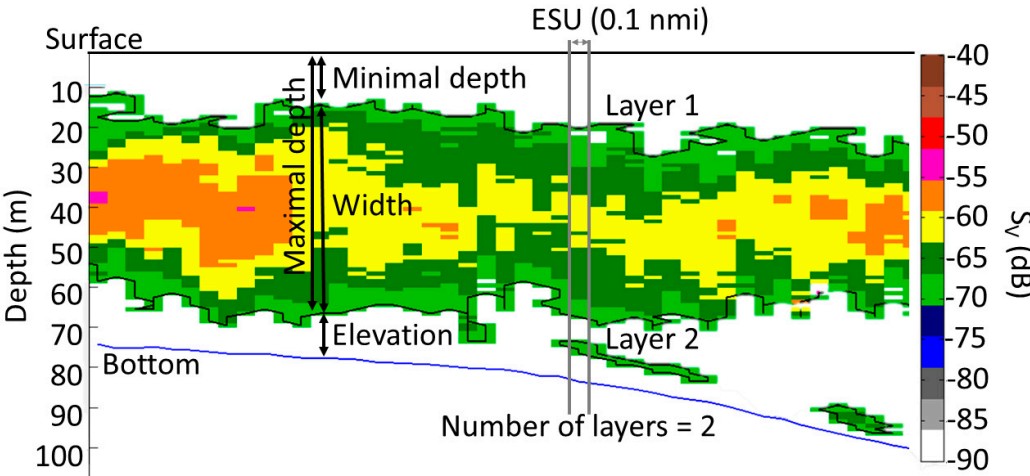

**Figure 2.** Schema showing typical sound-scattering layers (SSLs) variables exported under Matecho. Maximum depth of shallowest SSL (Đ1 in m), minimum depth of shallowest SSL (đ1 in m), vertical width of shallowest SSL (Ẇ1 in m), elevation above the sea floor of shallowest SSL (Å1 in m) and total number of SSLs (Ꞑ) present (here, n = 2) in the water column. The right axis depicts mean volume backscattering strength (Sv in dB) and left axis depicts local bottom depth. SSLs were extracted and contoured by a black curve. Black bold line represents sea surface; the sea bottom is represented by a blue line. A single elementary sampling unit (ESU) of 0.1 nmi is shown as example, outlined by two vertical grey lines.

From this set of variables, we inferred new variables: the elevation of the shallowest SSL ($\mathring{A}_{1,j}$) and all SSLs ($\mathring{A}_{all,j}$) [62]; the proportion of water column occupied by the shallowest SSL ($\dot{P}_{1,j}$) and by all SSLs ($\dot{P}_{all,j}$); and the relative importance of the shallowest SSL compared to all SSLs ($\dot{C}_{1,j}$). All calculations are detailed in Table 2. In total, we worked with 14 variables.

**Table 2.** List of all sound scattering layer (SSL) variables, symbol, unit and formulae. Sv is the volume backscattering coefficient in dB, sA is the area backscattering coefficient ($m^2$ $nmi^{-2}$) [6]. N/A means category is not applicable. The term "i" is the SSL number, starting at 1 for the shallowest SSL and called "all" when including all SSLs. The term "j" is the ESU number.

| Calculated Variable | Symbol | Unit | Formula | Reference(s) |
|---|---|---|---|---|
| Bottom depth at ESU j | $D_j$ | m | N/A | - |
| Number of SSL at ESU j | $N_j$ | - | N/A | [30,62,63] |
| Minimum depth of all SSLs at ESU j | $đ_{all,j}$ | m | N/A | [62,64] |
| Maximum depth of the shallowest SSL at ESU j | $Đ_{1,j}$ | m | N/A | Adapted from [62,64] |
| Maximum depth of all SSLs at ESU j | $Đ_{all,j}$ | m | $\sum\limits_{i=1}^{N} Đ_i$ | Adapted from [62] |
| Minimal elevation of the shallowest SSL at ESU j | $\mathring{A}_{1,j}$ | m | $D_j - Đ_{i,j}$ | Adapted from [62,65] |
| Minimal elevation of all SSLs at ESU j | $\mathring{A}_{all,j}$ | m | $D_j - Đ_{all,j}$ | Adapted from [62,65] |
| Width of shallowest SSL at ESU j | $\dot{W}_{1,j}$ | m | $Đ_{i,j} - đ_{i,j}$ | [28,62] |
| $S_v$ of shallowest SSL at ESU j | $S_{v,1,j}$ | dB re 1 $m^{-1}$ | $10 \log_{10} s_v$ | Adapted from [6] |
| Mean $S_v$ of all SSLs at ESU j | $S_{v,all,j}$ | dB re 1 $m^{-1}$ | $10 \log_{10} \dfrac{\sum_{i=1}^{N} 10^{s_{v,i,j}/10}}{N}$ | Adapted from [6] |
| $s_A$ of shallowest SSL at ESU j | $s_{A,1,j}$ | $m^2$ $nmi^{-2}$ | $4 \pi (1852)^2 s_a$ | Adapted from [6] |
| Mean $s_A$ of all SSLs at ESU j | $s_{A,all,j}$ | $m^2$ $nmi^{-2}$ | $\dfrac{\sum_{i=1}^{N} s_{A,i,j}}{N}$ | Adapted from [6] |
| Proportion of the water column occupied by the shallowest SSL at ESU j | $\dot{P}_{1,j}$ | % | $\dfrac{\dot{W}_j}{D_j}$ | Present study |
| Proportion of the water column occupied by all SSLs at ESU j | $\dot{P}_{all,j}$ | % | $\sum\limits_{i=1}^{n} \dot{P}_{i,j}$ | Adapted from [30] |
| Contribution of the shallowest SSL in the proportion of the water column occupied at ESU j | $\dot{C}_{1,j}$ | - | $\dfrac{\dot{P}_{i,j}}{\dot{P}_{all,j}}$ | Present study |

We used the same SSL variable set for all AA LMEs, allowing us to compare ecosystems and analyse their diel migrations and inter-annual variabilities. Two of the SSL variables (number of SSLs and proportion of the water column occupied) that we used had already been used by [30,63,66], whereas two others (proportion of the water column occupied by the shallowest SSL and contribution of shallowest SSL in the proportion of the water column occupied) are new ones. The mixed layers, minimum oxygen and euphotic zone depths should influence both variables involved in calculating proportional occupancy of the water column. Other variables used have traditionally characterized acoustic pelagic fish school using, e.g., "shoal echo-integration" [62] and adapted here to SSL.

We calculated all statistics using R version 3.4.3 [67]. We used boxplots to compare medians and data distributions of continuous variables associated with the four marine ecosystems. We used ANOVA and the Wilcoxon test to compare non-parametric distributions of data between paired LMEs. For the comparison of more than two LMEs, we tested the marine ecosystems by pair. Differences were considered significant if the probability (*p*-value) of the statistical test was < 0.05. We used kernel density estimates [68] to scrutinize differences in distributions identified by the Wilcoxon tests. The resolution of each variable was used to define bandwidth parameters in the kernel density function. For discrete variables, we drew barplots and applied Chi-square tests (*p*-value < 0.05) to examine the extent of independence between variables [69].

To analyse differences between day and night in the four marine ecosystems, we drew two boxplots for each variable, one for daylight conditions and one for night-time conditions. A Wilcoxon test was used to compare means between day and night for each ecosystem and each variable. For discrete variables, we used the Chi-square test ($p$-value < 0.05) to compare values between day and night. To estimate inter-annual change, we applied a linear regression ($p$-value < 0.05) incorporating the mean value for each variable for each year. Polynomial regressions were also tested up to order 3.

A principal component analysis (PCA) was run to summarise multi-dimensional information and identify correlations between variables. The package "FactoMineR" [70] was used to run the PCA with standardized data.

Results are organised to present (i) the spatial variability between marine ecosystems, (ii) differences in diel vertical migrations organisations and (iii) changes over years.

## 3. Results

### 3.1. Spatial Variability of Sound Scattering Layers

When we compared the northern and southern areas of CCLME, we found that measurements of the variables for these two marine ecosystems differed significantly from one another (Supplementary Material S1). At a larger scale, i.e., among all AA LMEs, we found that 83 ANOVA tests were significant among 84 and, which indicates that variables were significantly different from one another according to Wilcoxon test results (see Supplementary Material S1 for descriptors not presented here), except $s_{A, all}$ between CCLME North and GCLME.

3.1.1. Number of Sound Scattering Layers ($Ņ$) in the Water Column

The number of SSLs per ESU ($Ņ$) differed significantly among marine ecosystems (Figure 3). CCLME south and GCLME had the highest number of ESUs ($Ņ$) within a single SSL (more than 80%), whereas CCLME north and BCLME had the highest number of ESUs with either no SSLs ($Ņ = 0$) or more than one SSL ($Ņ > 1$). However, CCLME north and BCLME did not have the same mean number of SSLs. BCLME had more ESUs without any SSLs ($Ņ = 0$) than any other marine ecosystem, but it also had the lowest proportion of ESUs with a single SSL ($Ņ = 1$).

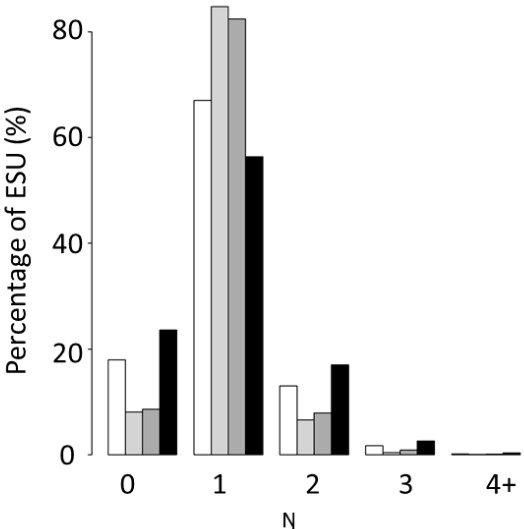

**Figure 3.** Barplots of number ($Ņ$) of sound-scattering layers (SSL) in percentage exhibited by four marine ecosystems off West Africa: the Canary Current Large Marine Ecosystem north (white) and south (in light grey), Guinea Current Large Marine Ecosystem (dark grey) and the Benguela Current Large Marine Ecosystem (black). ESU denotes elementary sampling units (0.1 nmi). The value 4+ includes ESUs with at least four SSL in the water column.

The analyses highlighted that ESUs with $N = 1$ are the majority whatever the marine ecosystems considered. In all marine ecosystems studied, $N \leq 6$. Six layers ($N = 6$) where found in 90 ESUs, usually in waters of 100–140 m bottom depth.

### 3.1.2. Minimum Depth of All SSLs ($đ_{all}$)

Minimum depths of all SSLs ($đ_{all}$) differed significantly among marine ecosystems (Figure 4a). Boxplots showed a strong difference in $đ_{all}$ between the northern and southern CCLME marine ecosystems (Figure 4). The northern part of CCLME presented more important values of median and standard error ($15 \pm 16$ m) for this variable than did the southern part ($10 \pm 7$ m). However, density peaks of minimal depth of SSLs of the marine ecosystems occurred at similar depths (Figure 4b). Only size of the peak differed. Boxplots and density curves for the BCLME were similar to those of the northern CCLME, whereas data for GCLME were similar to the southern CCLME. The data showed that the four marine ecosystems conformed to two basic types: one type with a distinctive, high-density peak (CCLME south and CGLME), and a second type with a lower, less distinct density peak (CCLME north and BCLME).

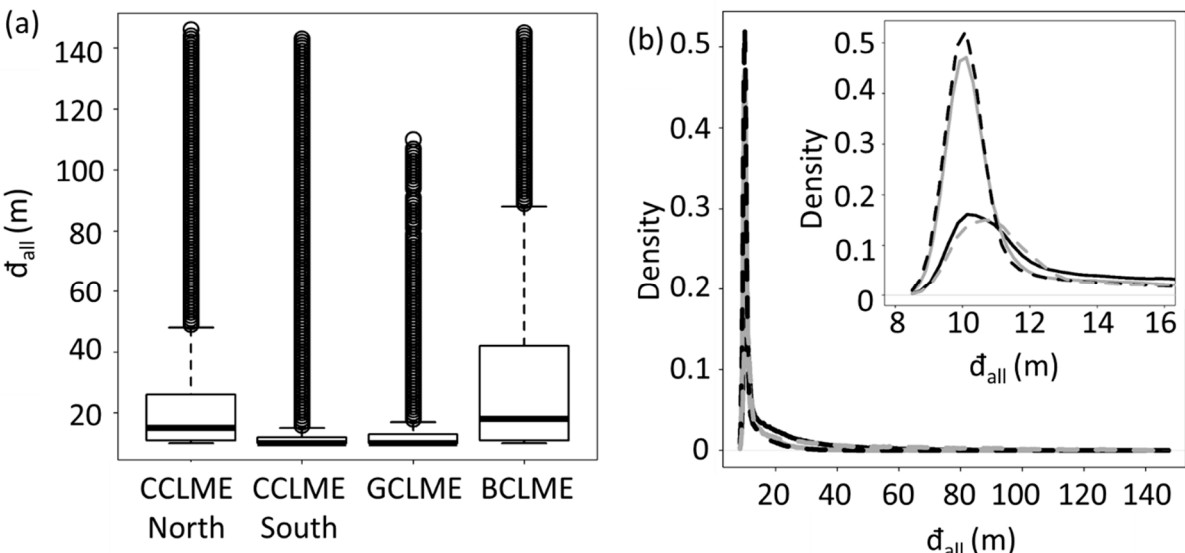

**Figure 4.** Variations in means and densities of minimal depth of sound-scattering layers (đall) for four marine ecosystems off West Africa. Panels for the northern Canary Current Large Marine Ecosystem (CCLME), southern CCLME, Guinea Current Large Marine Ecosystem (GCLME) and Benguela Current Large Marine Ecosystem (BCLME): Panels: (**a**) boxplots showing the median, first and third quartiles (central box) of minimum depth (đall), with external lines representing data range (the upper point is the maximum or the sum of the third quartile and 1.5 times the amplitude between first and third quartile, whereas other points are outliers) and (**b**) density curves for minimum depth (đall) for CCLME north (full black line), CCLME south (dotted black line), GCLME (full grey line) and BCLME (dotted grey line). The top right of panel (**b**) is a blow-up of the density curves at the lower left over a depth range from 8 to 16 m.

### 3.1.3. Proportion of Water Column Occupied ($\dot{P}_{all}$)

The proportion (in volume) of the water column occupied by all SSLs ($\dot{P}_{all}$) differed significantly among marine ecosystems; boxplots highlighted variations for these differences (Figure 5a). Density curves for SSLs were bimodal and the peak density values differed between the northern and southern CCLMEs (Figure 5b). The highest peak (on the right side of Figure 5b, for $\dot{P}_{all}$ close to 100%), showed that for most ESUs (except in BCLME), the water column was filled more than 90% by SSLs. CCLME north and the BCLME were similar to each other and much different from the other two marine ecosystems relative to

the proportion of the water column (by volume) occupied by SSLs. Indeed, only BCLME (and northern CCLME to a lesser extent) showed a density peak representing a sparsely filled water column ($\dot{P}_{all}$ close to 0–10%).

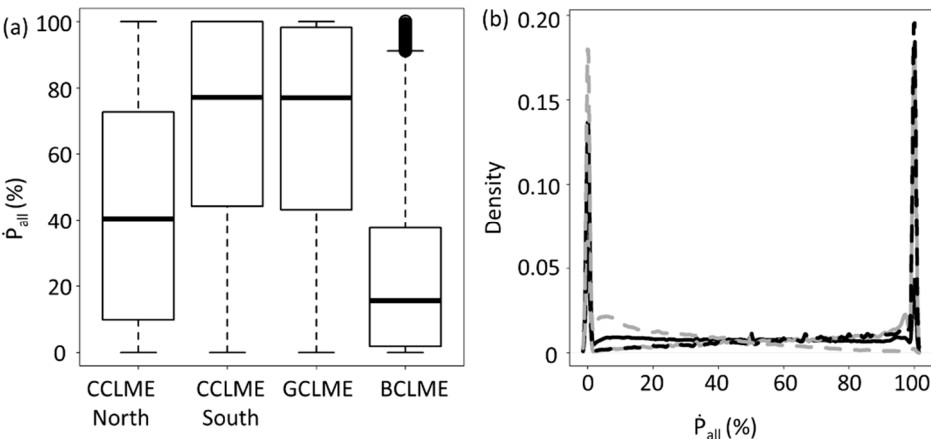

**Figure 5.** Variations in percent of the water column occupied by all sound-scattering layers for four marine ecosystems off West Africa. Panels for the northern Canary Current Large Marine Ecosystem (CCLME), southern CCLME, Guinea Current Large Marine Ecosystem (GCLME) and Benguela Current Large Marine Ecosystem (BCLME). Panels: (**a**) boxplots depicting the median, first and third quartiles (central box) of percent of water column occupied by all sound-scattering layers ($\dot{P}all$), with external lines representing data range (the upper is the maximum or the sum of the third quartile and 1.5 times the amplitude between first and third quartile, whereas the other points are outliers). (**b**) Density curves for percent of water column occupied by all sound-scattering layers ($\dot{P}all$) in CCLME north (full black line), CCLME south (dotted black line), GCLME (full grey line) and BCLME (dotted grey line).

3.1.4. Mean Volume Backscattering Strength of First Sound Scattering Layer ($S_{v, 1}$)

$S_{v, 1}$ differed significantly among studied marine ecosystems (Figure 6a). Density curves of all marine ecosystems showed similar shapes, but they peaked at slightly different values (about −65.0/−60.0 dB). The peak backscattering strength for CCLME north was a bit lower than for CCLME south. The peak for CCLME north, at −65.6 dB, is close to the peak for BCLME (−66.6 dB), whereas the peak of CCLME south (−63.5 dB,) is similar to the peak for GCLME (−64.0 dB). However, mean $S_{v, 1}$ values are quite similar among marine ecosystems.

*3.2. Comparative Analysis of Diel Differences among Marine Ecosystems*

All pelagic marine ecosystems in this study differed in their SSL spatial organisation over a diel cycle. All variables measured among marine ecosystems during daytime and night-time differed significantly (Figure 7), except for the maximum depth of the shallowest SSL ($Đ_1$) in BCLME (other variables, whose $Đ_1$, are shown in Supplementary Material S2 as statistical tests).

In all four marine ecosystems, $đ_{all}$ (minimum depth of all SSLs) was deeper during the day than at night and included a more extensive range of depths (Figure 8a). The water column was also more filled with organisms at night than during the day. $S_{v, 1}$ (mean volume backscattering strength of the shallowest SSL) was higher at night, except for water mass BCLME, where $S_{v, 1}$ was similar during day and night. The number of ESUs ($ N $) without SSLs was lower for all ecosystems at night. In CCLME north and BCLME, the proportion of ESUs with four SSLs or more was higher at night.

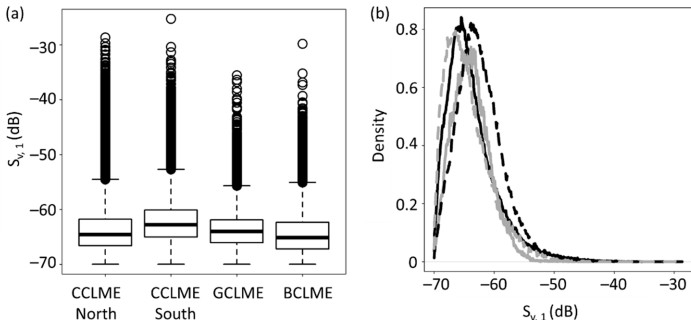

**Figure 6.** Variations in backscattering strength of the shallowest sound-scattering layers (Sv, 1) of four marine ecosystems off northern Canary Current Large Marine Ecosystem (CCLME), southern CCLME, Guinea Current Large Marine Ecosystem (GCLME) and Benguela Current Large Marine Ecosystem (BCLME). Panels: (**a**) boxplots showing the median, first and third quartiles (central box) of backscattering strength of the shallowest SSL (Sv, 1), with external lines representing data range (the upper is the maximum or the sum of the third quartile and 1.5 times the amplitude between first and third quartile, whereas the other points are outliers). (**b**) Density curves for backscattering strength of the shallowest SSL (Sv, 1) in CCLME north (full black line), CCLME south (dotted black line), GCLME) (full grey line) and BCLME (dotted grey line).

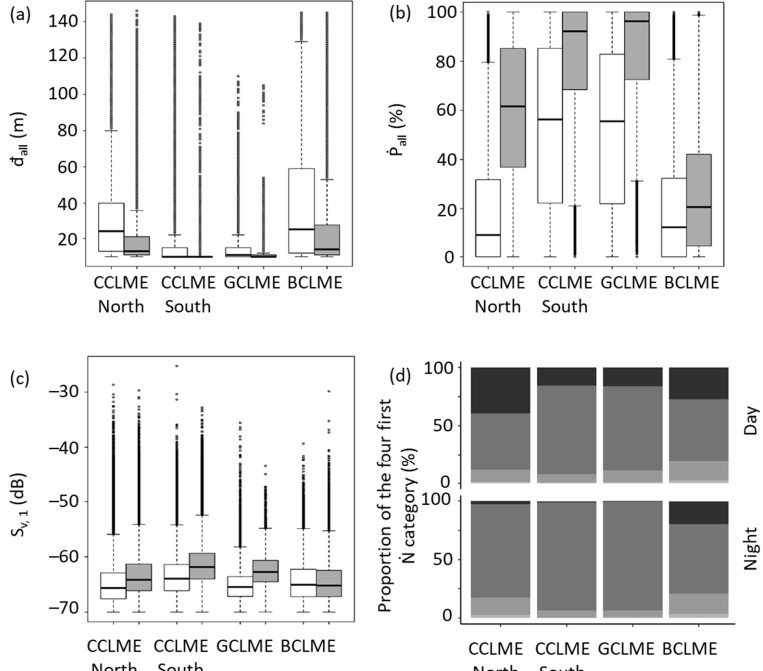

**Figure 7.** Variations in characteristics of sound-scattering layers (SSLs) from four marine ecosystems off West Africa. Panels for the Canary Current Large Marine Ecosystem North (CCLME north) and South (CCLME south), Guinea Current Large Marine Ecosystem (GCLME) and Benguela Current Large Marine Ecosystem. Panels: (**a**) boxplots of daytime (white) and at night (grey) depicting minimum depth of shallowest SSL ($d_{all}$). (**b**) Boxplots of daytime (white) and at night (grey) of the proportion of the water column occupied by all SSLs ($\dot{P}_{all}$). (**c**) Boxplots of daytime (white) and at night (grey) of mean volume backscattering strength of shallowest SSL (Sv, 1). (**d**) Bar chart for proportion ($\dot{N}$) of SSLs in an elemental sampling unit (black = no SSL; dark grey = one SSL; grey = two SSLs; light grey = three SSLs; very light grey = four SSLs or more). Boxplots depict the median, first and third quartiles in the central box. External lines represented data range (the upper point is the maximum or the sum of the third quartile and 1.5 times the amplitude between first and third quartiles). Other points are outliers.

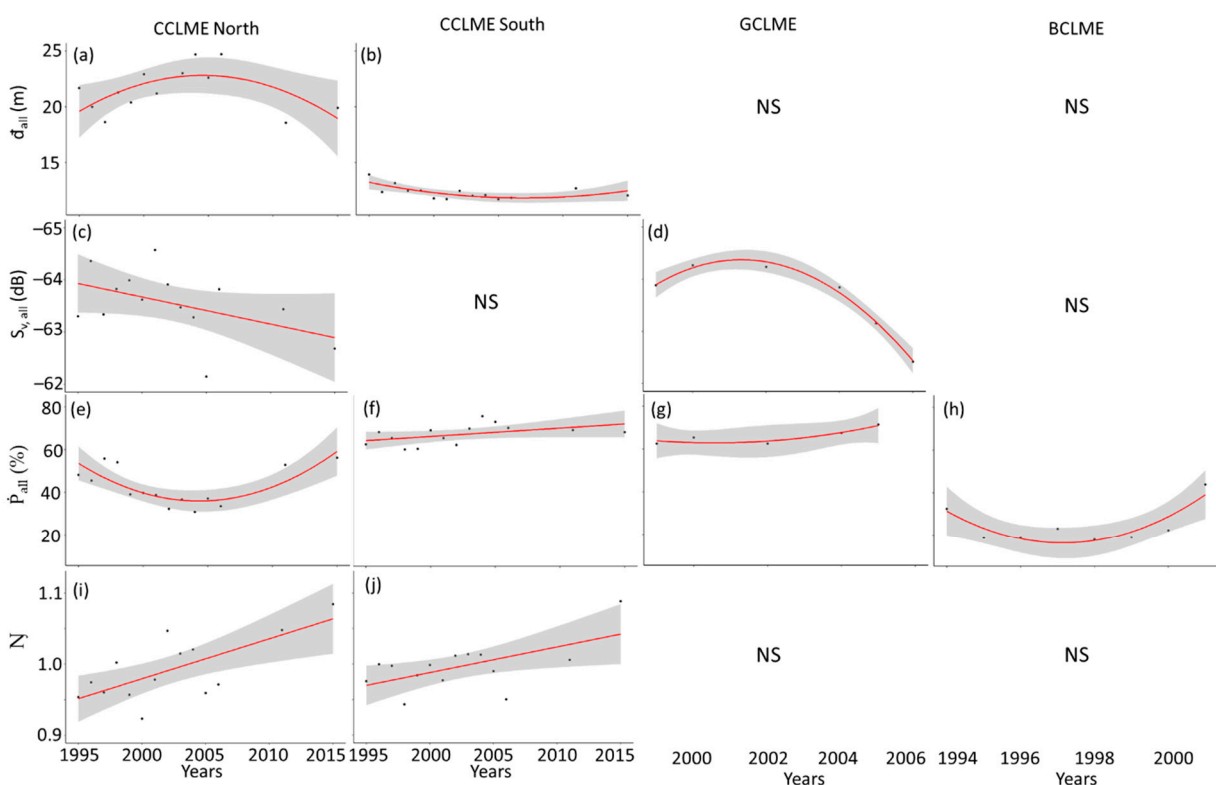

**Figure 8.** Selected significant variables for sound-scattering layers (SSLs) of four marine ecosystems off West Africa. Panels for (**a**,**c**,**e**,**i**) the northern Canary Current Large Marine Ecosystem (CCLME), (**b**,**f**,**j**) southern CCLME, (**d**,**g**) Guinea Current Large Marine Ecosystem (GCLME) and (**h**) Benguela Current Large Marine Ecosystem (BCLME). Panels present mean by year and significant trend for (**a**,**b**) minimal depth (đ1) in m; (**c**,**d**) Mean volume backscattering strength of all sound-scattering layers (SSLs) (Sv, all) in dB; (**e**–**h**) Proportion of the water column occupied by all SSLs (Ṗall) in %; (**i**,**j**) Number of SSLs (Ꞑ). Non-significant regressions have not been plotted.

### 3.3. Inter-Annual Variability and Trends in the Atlantic African Large Marine Ecosystems

Testing variables to detect annual changes (1995–2015) in each marine ecosystems independently (Figure 8, Table 3), we found few SSL variables significantly responding. We report two types of inter-annual variations: without or with trend over years.

For the northern CCLME, the mean volume backscattering strength of all SSLs ($S_{v,\,all}$) and the number of SSLs (Ꞑ) were the only two variables which decreased and increased, respectively, over the years (Figure 8c,i). The same variable (Ꞑ) significantly increased over years in the southern CCLME (Figure 8j). In the southern CCLME, there was a significant increase in the proportion of the water column occupied by all SSL ($\dot{P}_{all}$). The GCLME possessed three variables that showed significant changes over years: the proportion of the water column occupied by the shallowest SSL ($\dot{P}_1$), the proportion occupied by all SSLs ($\dot{P}_{all}$) (Figure 8f), and the proportion of the water column occupied by the shallowest SSL relative to the proportion occupied by all SSLs (the ratio $\dot{C}_1$). All the above-described variables increased over the time period examined. BCLME did not show any significant trend in variables over the period examined. All significant regressions are presented in Supplementary Material S3.

Regressions were initially tested using the whole dataset, but only means by years have been graphically represented. Significant variables using means by year were also significant using averaged data.

**Table 3.** Summary of the significance tests of the linear regression for change over time (y) for sound-scattering layer (SSL) variables collected from African Atlantic Large Marine Ecosystems. "NS" represents a non-significant $p$-values > 0.05. In each cell, the order of the polynomial regression and the adjusted $R$-squared ($R^2$) are written, followed by the associated $p$-value in brackets.

| Sound-Scattering Layers Variables (SSL) | CCLME | | GCLME | BCLME |
|---|---|---|---|---|
| | **North** | **South** | | |
| Minimum depth of all SSLs ($đ_{all}$) | $R^2 = 7.45 \times 10^{-3}$ ($p < 2.2 \times 10^{-16}$) | $R^2 = 4.56 \times 10^{-3}$ ($p < 2.2 \times 10^{-16}$) | NS | NS |
| Maximum depth of first SSL ($Đ_1$) | $R^2 = 3.46 \times 10^{-3}$ ($p < 2.2 \times 10^{-16}$) | NS | NS | NS |
| Width of first SSL ($Ẇ_1$) | $R^2 = 1.35 \times 10^{-2}$ ($p < 2.2 \times 10^{-16}$) | NS | NS | NS |
| Minimum elevation of first SSL ($Å_1$) | NS | $R^2 = 7.19 \times 10^{-3}$ ($p < 2.2 \times 10^{-16}$) | NS | NS |
| Minimum elevation of all SSLs ($Å_{all}$) | NS | $R^2 = 5.11 \times 10^{-3}$ ($p < 2.2 \times 10^{-16}$) | NS | NS |
| Mean volume backscattering strength of first SSL ($S_{v,1}$) | NS | NS | $R^2 = 2.71 \times 10^{-2}$ ($p < 2.2 \times 10^{-16}$) | NS |
| Mean volume backscattering strength of all SSLs ($S_{v,all}$) | $R^2 = 4.73 \times 10^{-3}$ ($p = 0.090$) | NS | $R^2 = 2.76 \times 10^{-2}$ ($p < 2.2 \times 10^{-16}$) | NS |
| Nautical area scattering strength of first SSL ($s_{A,1}$) | NS | NS | NS | NS |
| Mean nautical area scattering strength of all SSLs ($s_{A,all}$) | NS | NS | NS | NS |
| Proportion of the water column occupied by first SSL ($Ṗ_1$) | $R^2 = 4.60 \times 10^{-2}$ ($p < 2.2 \times 10^{-16}$) | NS | $R^2 = 2.37 \times 10^{-2}$ ($p < 2.2 \times 10^{-16}$) | $R^2 = 6.64 \times 10^{-2}$ ($p < 2.2 \times 10^{-16}$) |
| Proportion of the water column occupied by all SSLs ($Ṗ_{all}$) | $R^2 = 4.45 \times 10^{-2}$ ($p < 2.2 \times 10^{-16}$) | $R^2 = 4.22 \times 10^{-3}$ ($p = 0.084$) | $R^2 = 1.81 \times 10^{-2}$ ($p < 2.2 \times 10^{-16}$) | $R^2 = 7.18 \times 10^{-2}$ ($p < 2.2 \times 10^{-16}$) |
| Contribution of first SSL in the proportion of the water column occupied by ($Ċ_1$) | $R^2 = 8.43 \times 10^{-3}$ ($p < 2.2 \times 10^{-16}$) | NS | $R^2 = 3.80 \times 10^{-3}$ ($p < 2.2 \times 10^{-16}$) | NS |
| Number of SSLs ($Ŋ$) | $R^2 = 2.55 \times 10^{-3}$ ($p = 4.83 \times 10^{-3}$) | $R^2 = 2.37 \times 10^{-3}$ ($p = 0.0242$) | NS | NS |

### 3.4. Global Analyse of Correlations between SSL Variables

The two first dimensions of the PCA captured 49% of the variability (Figure 9). Based on the vectors of the PCA diagram, we classified variables into four groups: (1) $Ŋ$, $Å_1$ and $Å_{all}$, (2) $Đ_{all}$, $Đ_1$ and $Ẇ_1$, (3) $Ċ_1$, $Ṗ_1$ and $Ṗ_{all}$ and (4) all acoustic variables. The variable $đ_{all}$ is located between group 1 and 2, with low correlation with other variables.

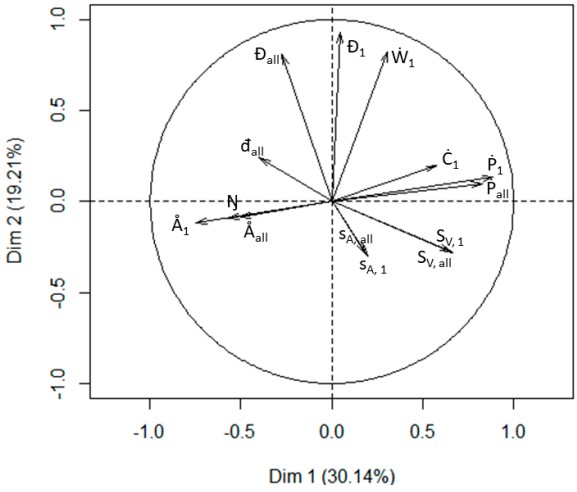

**Figure 9.** Principal component analysis diagram derived from survey data collected from the Atlantic African Large Marine Ecosystems including all the 14 SSL (Sound-scattering Layers) variables.

## 4. Discussion

### 4.1. Relevance of Choice of Variables

SSL variables provided insights into spatio-temporal distributions of mid-trophic pelagic organisms. Some variables appeared to be correlated with one other (e.g., $\dot{P}_1$ with $\dot{P}_{all}$ and $S_{v,1}$ with $S_{v,all}$), which suggests a high importance of the shallowest SSL in the four marine ecosystems studied. Indeed, this SSL drove a high part of the variability and there was often a single SSL in the water column. The PCA analysis gave a general idea of correlations and groupings within the data. This PCA captured 49% of the variability, which highlighted the diversity of variables. The two main axes were not able to capture the diversity. However, although all of the variables we examined were useful in revealing distributions of pelagic organisms, other SSL variables might prove more important in other marine ecosystems or were the ecosystems studied to become altered. Innovative variables presented here were found to be efficient to monitor the ecosystems, and especially to detect trends over time, e.g., the number of SSLs. An ecosystem can be monitored with fewer variables by deleting some of the variables that auto-correlate, but one must be careful to retain some variables capable of detecting ecosystem change, i.e., some variables may become more important than others in altered systems. In any case, the set of variables that we used in this study appeared to be capable of distinguishing differences both within and between marine pelagic ecosystems. Indeed, the marine ecosystem studied are already known to exhibit different system organisation [51]. SSL variables appeared efficient to characterize on various and heterogeneous marine pelagic ecosystems even if future application on other case studies would require specific adaptations.

### 4.2. Comparative Analysis of Sound-Scattering-Layer Variables within and between Atlantic African Large Marine Ecosystems

SSL variables were used to compare marine ecosystems inside LME, such as CCLME North and South, and between AA-LMEs. As expected, we found significant differences, which can be explained by specific processes already described in the literature.

Previous studies explored differences between SSLs from northern and southern CCLME [34], which could be explained by the upwelling effect. In fact, upwelling in both the southern Canary Current LME and the Guinea Current LME were seasonal [34,54–56], whereas the upwelling in the northern Canary Current LME and the Benguela Current LME occurred year-round [53]. These differences were captured by our methodology, which highlights its efficiency. Upwelling involves many changes in biocenoses, especially in abundance and species occurrence. This phenomenon has been studied for plankton in all the AA LME [32,35,38]. Thus, the differences in functioning within the CCLME are well documented (e.g., [34,71]) and especially the difference between North and South parts [31,40]. In our results, GCLME function was close to the southern CCLME one. GCLME plankton was very sensitive to upwelling presence [72]. The upwelling regime of southern CCLME and GCLME lead micronekton populations in the same direction in both LMEs. BCLME, with functioning closer to northern CCLME, was quite different from other marine ecosystems. In fact, this area was characterised by extensive mixing and variability, which strongly influences the dynamics of the system [73]. SSLs were intrinsically linked to environmental parameters, and this high variability influences number and composition of micronekton population, which involves different water-column organisations.

Our SSL variables were influenced by the oceanographic conditions found in the different marine ecosystems and were therefore suitable to discriminate different marine ecosystems. For example, CCLME south and GCLME characterized by seasonal upwelling had the highest number of ESUs with only one SSL, while CCLME north and BCLME, where the upwelling is permanent, had the highest number of ESUs with no SSLs. The same pattern has been observed for $S_{v,1}$ ($S_{v,1}$ peak for northern CCLME, and BCLME was a bit lower than CCLME South and GCLME). Indeed, SSLs need stable hydrological conditions to form and persist [74,75]. For instance, in Monterey Bay (California), a decline in acoustic

backscatter intensity was observed in the upper part of the water column immediately following an upwelling event [76].

Changes in the SSLs' elevation ($\mathring{A}_{all}$, distance between the deepest SSL and the sea floor) were not explained by the location of the euphotic zone because changes are also found in shallow parts of the shelf. Thus, the height of a SSL above the sea floor is likely related to physiochemical phenomena (e.g., water turbulence, internal wave effects on the bottom, or the existence of a bottom layer [77] possessing different physiochemical characteristics than the overlying water column). An alternative explanation for changing heights of SSLs above the sea floor could be due to the behaviour of pelagic mid-tropic organisms, linked or not [78] with physical or environmental characteristics. Variations might also be explained by the high biodiversity of the AA LME [79,80], especially in light of the intense upwelling area near latitude 26–27° South, just off Lüderitz Bay [39,73]. All zooplankton systems within Atlantic African LME are dominated by copepods [81]; phytoplankton are dominated by diatoms and there are few endemic species. Species composition can largely change over season and years linked to environmental parameters. Therefore, we speculate that the differences we observed among LMEs are likely due to physical and biogeochemical characteristics of the pelagic habitat [72].

All variations detected by SSL variables have a biological reality and an explanation previously studied. This highlights the efficiency of our method to monitor ecosystem and capture their variability using SSL variables.

### 4.3. Comparison of Diel Vertical Migration

The SSL variables, efficient for comparing marine ecosystems, also are useful to perform comparisons within the same ecosystem, such as variation in diel organisations. Therefore, SSLs were more distributed throughout the water column at night than during the day, with a higher proportion of the water column occupied by organisms (and wider SSLs) at night in the whole CCLME. In addition, SSLs were closer to the surface during the night, a characteristic typical of nocturnal ascendant diel vertical migration (DVM) behaviour [82]. DVM is a behavioural mechanism generally characterized by an ascent during the night to feed and a descent to avoid predation by visual predators during the day [22,59]. Higher acoustic densities ($S_v$ and $s_A$) at night than during the day in the CCLME and GCLME can be explained by the diel migration, which involved populations with different aggregative behaviour between day and night. This difference of acoustic density could also be explained by the behaviour of some species, which could be too close to bottom or to surface during the day to be detected by acoustic tools and to be visible in the water column only during the night. In contrast, the BCLME exhibited different DVM behaviour, with similar values for acoustic variables during day and night and no difference in the maximum depth of the shallowest SSL, which could be explained by other populations, who present different diel behaviour. Moreover, the proportion of ESUs without SSL was much higher at night in the BCLME than in the other LME systems that we examined. We conclude that the mid-trophic compartment is more scattered throughout the water column at night than during the day. Therefore, we assume that the difference in the DVM movements in the BCLME were due to predator–prey interactions rather than physical or biogeochemical differences. Indeed, our results showed that during daytime, SSL variables were similar to those recorded in the northern CCLME.

### 4.4. Annual Variability and Trends in Atlantic African Large Marine Ecosystems

SSLs are known to present high variability over years [35]. This variability is known to be linked with some environmental parameters, such as sea surface temperature (SST) [35] or wind stress [38]. We find this variability in several regressions of second or third order, which did not present a trend, but highlights high inter-annual variability.

Moreover, we also observed trends in our study, which are out of usual inter-annual variability. Indeed, CCLME north had the highest number of ESUs, with no SSLs, whereas CCLME south had the highest number of ESUs, with only one SSL. These differences in SSL

number observed between the north and the south could be explained by the differences in the upwelling regime between both areas. Indeed, the CCLME north is characterised by a permanent upwelling that remains intense throughout the year [56], whereas the CCLME south is characterised by a seasonal upwelling in winter [53]. SSLs formation requires stable conditions in the water column [31]. SSLs form and persist under stable conditions allowing stratification (thermocline, pycnocline, halocline), i.e., in the absence of turbulence, caused by intense upwelling [83,84]. Our observations are also consistent with those of other studies highlighting trends, for environmental parameters, such as SST [31,85], and in some fishes and plankton populations [31,38,72]. In the mid-trophic compartment, the northern and southern CCLME were therefore impacted in the same way over the study period. Indeed, data suggest a substantial increase in upwelling intensity off northwest Africa [86–88], which could explain the changing distributions of organisms in the water column. Furthermore, Sarré et al. [52] reported during the same study period northward shifts in the distribution of *Sardinella* and other small pelagic species in the CCLME. This change in distribution has been attributed to low thermal tolerance species to high warming trends in the southern part of CCLME. The number of SSLs ($\mathbb{N}$) increased during the study period in both parts of CCLME. In the northern part, we also noticed an increase in $S_{v, all}$. In the southern part of CCLME, the filling rate of the water column is increasing. These trends can be attribute to the increase in upwelling intensity.

Three variables increased significantly in the GCLME ($\dot{P}_1$, $\dot{P}_{all}$, $\dot{C}_1$) during the study period and all were linked to the proportion of the water column occupied by the SSLs. This result highlights an increase in the proportion of the occupied water column, especially by the shallowest SSL. The growing importance of this SSL can be explained by an environmental change, favourable to particular pelagic species living in the upper part of the water column.

There are a few possible hypothesis for there being no change in SSL distributions in the BCLME: (1) there were no ecosystem changes during the decadal study period, (2) the SSL variables were not able to detect change, or (3) the BCLME was not spatially homogeneous and should have been partitioned into two or more homogeneous systems [79,89]. Indeed in the BCLME, the northern and southern area [52] were difficult to partition due to our sampling design. Moreover, there is few minor seasonal upwelling cells in the northern Benguela [73,79], and a high heterogeneity (different currents and natural borders), relating different function [79]. This phenomenon could require a special survey design.

The water column was inhabited by both vertically migrating and non-migrating species during the transition periods, which were not included in this study and would, if their compositions were known, provide important information regarding the structure and species composition of the ecosystems represented by the SSLs. Indeed, in absence of taxonomic data, which is often non-existent in sub-systems of the AA LMEs, the insights that we obtained on the temporal and spatial behaviours of the DVM still allowed us differentiate migrating from non-migrating species [90]. Future work should focus on examining alterations in DVM patterns as indicators of ecosystem change. Such information is needed for better understanding the role of micronekton and macrozooplankton on the biological pump [14,91].

## 5. Conclusions

In the AA LME, surveys covered a large area, with an important temporal range, which allow us to compare systems and to monitor them. The method developed here allows us to synthetically describe the pelagic component of these ecosystems and monitor them without additional data, e.g., biological sampling. Innovative descriptors, such as number of SSL or proportion of the water column occupied by shallowest SSL, complete existing SSL variables [6,28,30,48,62,63]. Half (7/14) of the variables presented in this work can be extracted directly by the Matecho open software.

The structures of SSLs are dynamic from diel to annual temporal scales and could be associated with specific water bodies as shown in this study. Even when species

composition appears to be stable within a system, the acoustic relative biomass of the SSLs temporally and spatially vary quantitatively [92]. It is challenging, time-consuming and often difficult to obtain an integrated view of an ecosystem including SSL roles and effects. However, SSL variables can provide composite and synthetic indices of pelagic conditions. Where the availability of data is "poor", or when logistical means are not available, a standardized method for collecting information on SSL provides data relevant for obtaining elementary information of the organisation of pelagic organisms over time. This method can monitor the functioning of the pelagic habitats over the continental shelf as in the high sea [48]. Such a method can be used to monitor changes in particular processes, e.g., changes in SSLs during DVMs [93]. Future work should concentrate on fine-scale processes [31,94] to provide new perspectives on interactions among indicators associated with processes occurring at the mixed layer depth, oxygen minimum zone, and other indicators associated with vertical structure in the water column, e.g., peak fluorescence, pycnocline [31]. A common spatial trait relative to all the systems we studied was the importance of ESUs with a single SSL compared to ESU with two or more SSLs. That is, the shallowest SSL is often unique in the water column and if not, it stays the most important, i.e., bigger and with higher acoustic density. The SSLs over the shelf were rarely multiple, but sometimes can reach six. In this work, the limitation to the shallowest SSL should not hide the interest of this new SSL variable to be applied in the high sea, where multiple layers often occur.

We also report the existence of a singular "empty layer" between the sea floor and the deepest part of the deepest SSL. This phenomenon can likely be characterised by as a yet unknown process(es), certainly of bio-physical origin, underlying once again the interest of such new descriptors of the SSL in marine sciences.

**Supplementary Materials:** The following supporting information can be downloaded at: https://www.mdpi.com/article/10.3390/fishes7020086/s1, Supplementary Material S1: inter-Large Marine Ecosystem comparison; Supplementary Material S2: comparison between day and night; Supplementary Material S3: Inter-annual regressions. References [Supplementary S1–S3] were cited in Supplementary Materials.

**Author Contributions:** Conceptualization, P.B.; methodology, A.M. and P.B. and Y.P.; software, Y.P., P.B. and A.M.; formal analysis, A.M. and N.B.; investigation, U.U., S.E.A., M.A.J., A.S. and A.M.K.; resources, P.B.; data curation, A.M. and N.B.; writing—original draft preparation, A.M., P.B. and U.U.; writing—review and editing, A.M., P.B., Y.P., N.D., A.S. A.M.K. and E.F.; visualization, Y.P.; supervision, P.B.; funding acquisition, P.B., A.S., S.E.A. and A.M.K. All authors have read and agreed to the published version of the manuscript.

**Funding:** This work started within the AWA project "Ecosystem Approach to the management of fisheries and the marine environment in West African waters", project funded by IRD and the BMBF (grant 01DG12073E and 01DG12073B), www.awa.ird.fr (accessed on 15 November 2021) (SRFC: Sub Regional Fisheries Commission) and the PREFACE project funded by the European Commission's Seventh Framework Program (2007–2013) under Grant Agreement number 603521, https://preface.b.uib.no/ (accessed on 15 November 2021) and ended within the TriAtlas European project (Grant Agreement number 817578).

**Institutional Review Board Statement:** Not applicable.

**Data Availability Statement:** The data is not available that is strategic data for each country related to their natural resources. Contact the author.

**Acknowledgments:** We thank the Nansen project (FAO/IMR) Jens-Otto Krakstad and IMR Norway as FAO and Africans colleagues for data collections. We also thank ARED (Doctoral Research Allowances) from Brittany (France).

**Conflicts of Interest:** The authors declare no conflict of interest.

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
