# Peer review of "Applying Acoustic Scattering Layer Descriptors to Depict Mid-Trophic Pelagic Organisation: The Case of Atlantic African Large Marine Ecosystems Continental Shelf"

_fishes, doi:10.3390/fishes7020086_

Round 1
Reviewer 1 Report
The manuscript is much improved. Although the grammar needs work (see next comment), the work is described clearly, the results follow the methods, and the discussion is more focused.
The English grammar needs substantial work. I strongly suggest that a native English writer and/or the journal editors work through the grammar.
Specific comments:
- Line 30. Define ESU.
- Line 97. Delete “Diel Vertical Migrations” and parentheses around “DVM”. The authors have already defined DVM.
- Line 396. I am not requesting the authors change the variable name “width” as it is defined in the paper as the vertical extent of the layer. Usually, width is used to measure a horizontal characteristic of a layer, whereas height or thickness are associated with vertical characteristics. Again, I do not expect the authors to change their terminology. I mention it for future use.
- Line 784. Change “deferred” to “differed”.
- Figure 8. Some time series appear linear, others do not. The authors may want to apply other regressions and look at some measure of fit to see if other types of relationships are better than linear regressions.
- Line 1298. I do not understand what is meant by “strong dominance of ESU with a single SSL”? Do the authors mean one SSL (not the same SSL) dominated the vertical structure over all AALMEs?
- Lines 1309-1313. The two sentences contradict each other. The first states the variables appear to be capable within and between marine pelagic ecosystems, but the second states the variables are not universal. The authors have come up with a nice set of variables describing SSLs. I suggest being a bit more positive and state that they should be applied to other ecosystems to see if they are valuable beyond the AALMEs studies.
Reviewer 2 Report
I like the approach of this manuscript and it has the potential to be an interesting and important contribution, and overall, I think it is well written. The methods are clear and concise. The organization of the results is in accordance with the methods. I found no inconsistencies in the text and no miscalculations. Only in the Discussion did I find a paragraph that is not clear enough, but with a minor revision, it can be overcome.
Specific comments
It seems to me that the first paragraph of section 4.2 is not clear how it links, in a harmonious way, with the work done by you. I suppose it is aimed at contextualizing the difference or similarity between the different AA-LMEs, but it is difficult to read. I suggest that it be worded in such a way that it links more harmoniously with the subsequent paragraphs.
Author Response
Please see attachment

This manuscript is a resubmission of an earlier submission. The following is a list of the peer review reports and author responses from that submission.
Round 1
Reviewer 1 Report
I like the concept of this manuscript and it has the potential to be an interesting and important contribution, but it is not well written. The methods are confusing and not concise, and the organization of the results does not parallel the methods. There are inconsistencies in the text and what appear to be incorrect calculations. But, overall the flaws are not fatal and with substantial revision, can be overcome.
General comments:
- The background in the Introduction is not comprehensive and sometimes inappropriate. For example, on line 45 the authors state that the study of SSL began in the 1980s, when in fact the study of SSLs began in the 1940s, or before. The Stanton paper (ref #10) does not deal with SSLs; it deals with the theory of scattering from cylinders. The authors need to dig a bit deeper into the literature and find papers that study SSLs. For example, read Benoit-Bird et al. (2017) Limnology and Oceanography, 62: 2788-2798, doi: 10.1002/lno.10606 and references therein.
- The methods are a very confusing read. The text does not seem to match what was done. This section needs to be completely rewritten to be more concise.
- Nearly all of Section 4.2 is based on conjecture with no basis in the results of the paper. Upwelling, oceanographic, and biological justifications are provided where no data are shown or analyses given in the manuscript to support these contentions.
Specific comments:
- Line 610. I do not think reference #1 is correct. It seems that #1 and #2 are the same.
- Figure 1. I suggest annotating this figure with the three LMEs.
- Line 66. Reference Figure 1.
- Line 127. I am very surprised the data were restricted to <150 and on the continental shelf. The classic SSLs are in oceanic waters (depths >1000m) and are found at depths of 400-600 m.
- Line 97. Somewhere in this paragraph the authors need to cite Table 1.
- Line 151. Sv is not a proxy for biomass. It is a proxy for abundance. That is because biomass is proportional to the cube of length, whereas density and abundance are proportional to the square of length, and acoustic backscatter is proportional to the square of length (e.g., backscattering cross-sectional area, m^2). It is unfortunate that this is becoming pervasive in the literature.
- Line 152. The authors mix citation style. It seems that MacLennan et al. and Behagle et al. should get numbers.
- Paragraphs beginning on line 169. This is a very confusing paragraph and I do not understand most of it. Please just state what was done, not what could have been done but was not.
- Lines 170-171. If the number of variables changed based on the number of SSLs, then the authors have chosen the wrong variables. The variables should not change, but the values of those variables should be representative of those features that the authors want to study.
- Line 170. Somewhere in this paragraph the authors need to cite Table 2.
- Table 2 caption. Sv does not have units of m^-1; it has units of dB (which was correctly defined in the text).
- Table 2 caption. sA does not have units of m^2 m^-2; that would be subscript “a”. Subscript “A” indicates NASC. The authors need to reread MacLennan et al. 2002.
- Table 2. Mean Sv is calculated incorrectly. The mean of the logarithm is not the logarithm of the mean. The authors need to translate Sv to sv, take the mean, then translate back to Sv. (capital letters indicate dB; lower case letters indicate linear form).
- Table 2 and associated text. It seems that another index variable is needed to refer to ESU. If I understand correctly, each variable was calculated for each ESU, so every variable should have an index of “j”, and most should have the index “i” indicating SSL.
- Lines 232-234. Acoustic data expressed as dB are already log10 transformed. Were dB values log-transformed again?
- The organization of the results section does not parallel the methods. The PCA gets a very small percentage of the methods, yet is the first topic in the results. The reader is expecting results of the individual variables first, then the PCA. The methods and results must parallel each other.
- Lines 252-253. Please explain how the groupings were done. For example, in my opinion, group 2 should include Dall, D1, and W1, but not d1.
- I am completely lost as to why only the shallowest layer was used. It seems the authors are greatly diminishing the power of their data by using only the shallowest layer. If this is the case, why limit the data to only the continental shelf and depths <150 m? Unless, the reason to use the shallowest layer is an artefact of limiting the data to <150 and on the continental shelf. The title of the paper should reflect only the shallowest scattering layer.
- If only the shallowest layer was used, Table 2 needs to be redone and any mention of other layers can be removed.
- Section 3.2. I thought all data in upwelling conditions were not used in the analyses? If so, with what data was this analysis done?
- Section 3.2.3. I would argue that the Sv values are quite similar among LMEs. Only 3 dB separate the mean values.
- Section 3.2.4. It seems that this should be the first section discussed. It shows that about 80% of the data had 0 or 1 layer. This could then be used as justification for selecting the shallowest layer. As the manuscript is written, the logic is backwards.
- Figure 9. The left graphs should include spaces for the missing years as do the right graphs.
- Table 3. I suggest either replacing the ns and * with the actual p values in the table, or convert this table to a graph where the significance is pictorially represented. A table should be reserved for actual values, whereas a graph can be used to show the overall results.
- Section 4.2. I thought upwelling data were not used in the analyses? If so, where does this section come from?
- Paragraph starting on line 449. Where is the analysis that shows the variables selected by the authors are strongly influenced by oceanographic conditions?
Reviewer 2 Report
Dear authors,
I am a little confused as to what the novelty of your study is. You write that it is the number of SSLs and the volume of water column occupied by the SSLs, but I think this is already used in common acoustic studies. Also, the role of acoustic methods and their ability to indicate a state and changes of the ecosystems are well known.